# Effectiveness of and Perspectives for the Sedimentation Analysis Method in Grain Quality Evaluation in Various Cereal Crops for Breeding Purposes

**DOI:** 10.3390/plants11131640

**Published:** 2022-06-21

**Authors:** Ilya Kibkalo

**Affiliations:** N.I. Vavilov All-Russian Institute of Plant Genetic Resources (VIR), 190000 St. Petersburg, Russia; i.kibkalo@vir.nw.ru

**Keywords:** wheat, rye, triticale, sorghum, maize, gluten, carbohydrate-amylase complex, SDS

## Abstract

The existing standardized methods for assessing the quality of marketable grain do not always meet the requirements of the breeding, such as the method’s rapidity, sufficiency of the minimum amount of experimental material, the minimal modifying influence of the external environment on the degree of expression of a criterion, and genetic determination and heritability of the latter. One of the methods that meets these requirements is the sediment volume test. The present study offers an analysis and examples of methodological developments in relation to the assessment of winter bread wheat grain in arid regions of cultivation, as well as of winter triticale. The fluorescent probing method was used as an example for demonstrating the prospects for assessing the swelling of ground grain products of both bread and durum wheat, and for such crops with a less-strong complex of storage polymers as triticale, rye, and millet. A two-stage sedimentation procedure that allows a successful differentiation of samples has been developed for sorghum and maize grain. It is presented here alongside with methodological works on wheat from different countries of the world. Examples of the proven high reproducibility of the sediment volume test in the offspring, and its genetic determination are provided. In general, the data obtained and the material accumulated by various researchers indicate that a modification of the sedimentation method, correctly chosen for specific goals and objectives, solves the problem of assessing grain quality in breeding samples starting from early progenies. All these circumstances make the sedimentation testing the leading or most promising method for assessing grain quality when breeding of a broad range of grain crops is carried out.

## 1. Introduction

Breeding for grain quality is the second key direction following breeding for productivity in the grain crops industry. The high quality of agricultural products, hence of food, can help guarantee the well-being and health of the population, and the food security of a country [1]. However, many analysts note that the share of high-quality grain in the gross harvest remains small in Russia [1,2]. The harvested grain quality is influenced by abiotic factors, depending on both the practices applied in agricultural production and the hereditary base of cultivated plants [3].

While the impact of agricultural practices of crop production on grain quality is standardized and limited, the genetic resources diversity, which constitutes the source of inheritable traits for crop breeding, offers the potential for agricultural development. The search for and selection of genotypes that have valuable properties remains an urgent task.

Due to historical and cultural peculiarities, the quality of cereal end products consumed by the population (bread, pasta, porridge) is a criterion that is defined with difficultly and depends on many technological factors such as technical processing, recipes, and conditions. Furthermore, the very concept of grain quality is described by a variety of methods and indicators, the number of which, despite the development of biotechnology, does not decrease, thus being an integral concept. Hence, there is a need for more elementary quality features, which can be determined in a standardized way, as experienced by the breeding, processing, and trading industries.

The testing of wheat (*Triticum aestivum* L. and *Triticum durum* Desf. L.) grain quality through the evaluation of dough rheological properties by using various instruments has received great international recognition. In breeding practice, the assessment of the genotypic diversity of wheat on the farinograph [4], alveograph [5,6], extensograph [7], mixograph [8], mixolab [9,10], and some other devices has become widespread. Such an analysis makes it possible to predict with great accuracy the baking value of accessions [11] and their suitability for making other dough products [12]. Rheological test parameters have been noted to be significantly determined genetically [13,14]. Moreover, the use of these devices makes it possible to predict the final product quality in the case of joint processing of wheat with other crops [15] and food components [16]. At the same time, it is noted that such tests are technologically complicated, and their results are subject to the influence of growing conditions and grain damage by pests [17]. In this regard, methodological work is carried out for modifying standard methods for breeding and genetic practice [18,19].

The amount of gluten and its quality determined by the IDK instrument (Russian abbreviation for the Gluten Deformation Meter) has become one of indicators for the assessment of wheat in the Russian Federation [20]. Indeed, the properties of gluten proteins and substances conglomerated with them determine to a significant extent the final product quality. However, this dependence is not always linear. For instance, it is known that the best bread in terms of volume is produced from the so-called filler wheat, i.e., medium-quality wheat [21]. With all the indisputable advantages of IDK as an indicator of gluten rheological properties [22,23], criticism of this criterion as a breeding character is also widespread. Indeed, as a rule, this indicator that differentiates the breeding material at a low level is subject to the influence of the conditions of this material cultivation and other factors, e.g., grain damage by *Eurygaster integriceps* (sunn pest). Hence, there have been attempts to modify the standard methodology for breeding needs [24]. Many researchers emphasize that strict IDK-based selection can lead to the loss of valuable genotypes, that it is more applicable to commercial grain consignments, and besides, it is not simple enough technologically [25,26], whereas the breeding practice needs a fast method that should employ minimum sample quantities of the tested material, reveal the qualitative potential of an accession, and the degree of expression of the method indicator should be determined to the maximum extent by the plant genotype.

The search for such a method has led to the appearance of the sedimentation analysis, i.e., a study of the ground wheat grain products swelling in water and weak acid solutions, which precisely indicates the quality of gluten and dough [27,28]. This analysis has gradually replaced the Berliner–Koopmann method of testing the swelling of gluten in weak acetic acid [29] and its modifications in Europe [30] and Russia [31,32], being much less laborious and more reproducible. Since its appearance [33], this method has undergone much evolution and adaptation to the needs of breeders in different countries and regions [34]. In different regions of the planet, micro-sedimentation assessment options were created, including those employing SDS, and recommended for determining grain quality for various types of wheat, for both *Triticum aestivum* L. [35] and *Triticum durum* Desf. L. [36,37,38,39].

It turned out that the place of wheat cultivation and breeding can significantly affect the efficiency of a particular modification of the method [40,41]. This is due to the colossal effect of climatic factors and varying weather conditions on the formation and conformation of protein structures of storage proteins, even in genotypes with a genetically fixed grain quality potential [3,42]. Therefore, methodological work related to this method is still underway, e.g., for adapting the SDS test for assessing the genetic diversity of hexaploid wheats in the USA [43] and Russia [44], including the detection of genotypic differences on the basis of the number of disulfide bonds in the gluten complex [24]; adaptation of the method for the evaluation of hard winter forms of *Triticum aestivum* L. [45]; for studying the structure of storage proteins in *Triticum aestivum* L. and *Triticum durum* Desf. L. with varying degrees of grain hardness [46].

## 2. The Influence of the Wheat Species and Conditions of Its Cultivation on the Effectiveness of Variants of Sedimentation Analysis

The weather conditions of the Middle and Lower Volga and other regions with frequent droughts promote the appearance of valuable hard-grain genotypes, which result from the breeding process and require a special methodological approach. For instance, in our experiments with flour sedimentation in 2% acetic acid, the old variety of spring wheat (*Triticum aestivum* L.) ‘Albidum 43’ which forms gluten of medium to weak strength and is often used as a filler wheat, significantly exceeded the sediment volume of the hard-grain variety ‘Saratovskaya 55’ that forms a strong gluten and can be an improver wheat in medium dry seasons. Only the extensive methodological work with SDS sedimentation made it possible to create optimal modifications of the test for bread [47] and durum (*Triticum durum* Desf. L.) wheat [48], which adequately differentiates breeding material in regions with hot and dry growing conditions [49].

The subsequent application of the method modification (created for spring bread wheat) to breeding material of winter bread wheat, showed its lower efficiency or an ambiguous relationship with traditional grain quality criteria [50,51], though it is used by some breeders for the differentiation of genotypes by grain quality [52]. The methodological work continued by the author on a total of 10 methods and their modifications using the material of winter bread wheat allowed the author to recommend his own version of the method for this crop (author’s previously unpublished data).

The study included the following sedimentation techniques:By Bebyakin and Buntina for spring bread wheat: a protocol by Bebyakin and Buntina, 1991 [47], two modifications according to the time of contact of the tested material with water, two modifications according to the time of sedimentation (SDS).By Bebyakin, Buntina, and Vasilchuk for spring durum wheat: a protocol by Bebyakin et al., 1987 [48], and two modifications according to the sedimentation time (SDS).By Knyaginichev, Komarov (70% yield of flour and acetic acid) [53].By Pumpyanskiy (1st grade flour and acetic acid) [54].

The study involved 16 varieties and lines of winter bread wheat harvested in 2008–2010. Due to the strong modifying effect of the digestive gland enzymes of sunn pest on the grain quality indicators, the studies were carried out both on the material cleared from the grains damaged by sunn pest, and on the unselected grain with a natural degree of damage by the pest, which amounted to 17–44% in 2008, 30–52% in 2009, and 13–39% in 2010.

It should be noted from the very beginning that, apparently, due to the fact that even the minimum percentage of damaged grains in the experiment was quite large and exceeded the possible content of defective grain with the maximum modifying effect, the degree of the grain mass damage by sunn pest did not significantly affect the results of studies of the unselected material within the yield of each year of the study.

The research results were subjected to ANOVA and correlation analysis.

It was found that all the studied techniques significantly (according to the F-criterion) differentiated the studied genotypes by the sediment volume in all the years of the study, except for the technique by Kniginichev–Komarov applied to the selected grain in 2009 [53]. Moreover, their results were mostly reliably related to each other (r = 64**–95**). However, their degree of differentiating ability, as well as the relationship with other grain quality indicators, was different.

Basically, the studied techniques in all their variations showed a medium differentiating capability (coefficient of variation V = 12–17). Only the Bebyakin–Buntina–Vasilchuk technique with modifications stood out; depending on the option, it could distinguish genotypes also at a high level (V = 17–35). This method was especially successful in differentiating the studied material naturally damaged by sunn pest, which gave a reason to talk about different genetic determination of accessions sensitivity to the pest infestation. For instance, in 2009 the varieties ‘Saratovskaya’ and ‘Sozvezdie’ showed a reduction in the sediment volume from the material naturally damaged by sunn pest by 60–65% in comparison with the samples taken from the damaged grain; the varieties ‘Smuglyanka’ and ‘Elvira’ demonstrated a 75–80% reduction, ‘Saratovskaya 8’ showed a reduction by 7% in 2008 and 2010, and other genotypes by 27–44%. Therefore, the growing conditions generally influenced the formation of the endosperm protein complex in the grain—the strength of the intra- and intermolecular bonds of protein macromolecules, which affected the sediment volume in general and, as a consequence, the degree of susceptibility to proteolytic enzymes of the pest. However, despite this, the samples with the genetically determined relative resistance of the protein complex to proteolytic enzymes were prominent against the general background in each of the studied generations. A similar phenomenon has been observed by other authors [55,56]. It has also been reported on the genetic variability of wheat in regard to the resistance to the proteolytic enzymes of the pest [57,58]. Thus, the sedimentation analysis of breeding material also allows the identification of genotypes with similar valuable properties.

The relationship between the sediment volume and the indicators of technological assessment of grain was forming in the following way.

In 2008, the sediment volume was reliably related to the IDK data (−0.56*…−0.71**; *, **-hereinafter the reliability of the correlation coefficient is at the 5 and 1% significance level, respectively), bread volume (0.51*–0.63**) and porosity (0.42–0.58*) only for the variants of the technique by Bebyakin–Buntina–Vasilchuk [48].

In 2009, sedimentation carried out by almost all the studied techniques and their modifications (except for that by Knyaginichev–Komarov) was reliably and practically to the same extent related to the rheological properties of dough measured by the alveograph (0.58*–0.71**); however, no statistically significant relation with the IDK and bread making quality data was found.

In 2010, the sediment volume turned out to be reliably associated with the IDK data (−0.49…−0.51*), bread volume (0.52*–0.61*), and porosity (0.49–0.53*) only for the Bebyakin–Buntina–Vasilchuk technique in its modification with the reduced sedimentation time.

Thus, the obtained data make it possible to recommend the Bebyakin–Buntina–Vasilchuk sedimentation technique with its modification for a reduced settling time for the express assessment of the winter bread wheat grain quality for breeding purposes, since it has the greatest informativeness and differentiating capability among all the studied techniques. Of interest is the use of this option for assessing both the grain damaged by sunn pest (naturally damaged, without selection) for identifying genotypes that are relatively resistant in terms of grain quality to the damage by this pest, and the selected grain for identifying genotypes that are potentially capable of forming a high-quality crop.

The values obtained using the proposed option differentiated accessions to a better degree, and were more closely and stably associated in years with other grain quality criteria. Thus, the differentiating ability (CV) of the IDK-1 (Gluten Deformation Meter) was 11.6–12.1%, the sediment volume determined by the technique accepted for the spring bread wheat was 12.0%, and 18.9–34.6% using the recommended technique. At the same time, the relationship of the said modification of the technique with the same IDK-1 data fluctuated over the years within the −0.51*…−0.71** range, while for the previous analogue it was not statistically significant in all years.

It is noteworthy that the modification of the method originally developed for durum wheat, but not for the spring bread wheat, turned out to be most successful for the winter bread wheat in the arid region. This is explained by the conditions of storage proteins formation in these crops. Due to its biology, winter wheat uses the naturally available moisture to a fuller extent, including the moisture in the autumn–spring period, thus behaving as a crop of intensive type, in a certain sense, with the biochemistry and physiology appropriate for a vegetating plant. This is the direction in which selection proceeds, that is, by identifying genotypes most responsive to favorable growing conditions. In arid regions, yield formation in spring bread wheat often occurs under conditions of moisture deficiency, therefore, it may be regarded as a conditionally extensive-type crop with the corresponding type of physiology and biochemistry. Resistance to unfavorable environmental factors is the direction in which selection is carried out. It is precisely these conditions that promote the emergence of hard-grain genotypes with an exceptional endosperm consistency. It is also known that bread and durum wheats have different types of “packing”, i.e., conglomeration of protein macromolecules [59]. The conglomeration of durum wheat storage proteins ensures a high minimum “grain hardness” and a smaller variability in the consistency of endosperm in general, while the conglomeration of bread wheat proteins allows for a high variability of the “packing density” of macromolecules, both towards a smaller and a larger degree. Due to the biological characteristics described above, winter bread wheat is most likely incapable of reaching the range of high endosperm density, and thus, in terms of the range of its maximum values, it coincides with that of durum wheat, yielding to it in protein conformation stability; hence, the reduced sedimentation time. Other researchers [46] also point out that the ranges of endosperm consistency partially overlap for certain groups of bread and durum wheat, which makes it possible to use the same methods in their assessment. These techniques are applied in practice, including modifications of the sedimentation test [60].

## 3. Application of Variants of the Sedimentation Analysis for Identifying Genetic Diversity by Grain Quality among Winter Triticale and Rye Accessions

The proposed modification turned out to be effective for winter triticale (X *Triticosecale* wittmack ex A. Camus). The difficulty of working with this crop is as follows: although its grain is formed as that of wheat and evaluated accordingly, from one of its ancestors, rye, it inherited a tendency towards an increased activity of grain enzymes, a relatively low protein content, often low protein quality, and a strong dependence on growing conditions, grain filling, and ripening.

However, the very first experience of using the SDS sedimentation test for the breeding material showed its high resolution capability and close correlation with the criteria of technological evaluation of grain [61,62], that is, with the IDK data (r = −0.71**), bread porosity (r = 0.69**), valorimetric number (r = 0.62**), dough development time (r = 0.64**), and dough softening (r = −0.45*) in farinograph units. Moreover, later it turned out that due to the above-described features of triticale quality formation, an adequate technological assessment of the breeding material was not possible each year because gluten could not be always washed (apparently due to the high proteinases activity), and it sharply decreased the differentiating capability of farinographic and bread-making quality assessment. It was only the SDS sediment volume that effectively differentiated the experimental material each year, somewhat varying in volume over the years. In previous works devoted to this method it has already been noted that it is relatively insensitive to the presence of defective grain in the sample, in particular, of that damaged by sunn pest [19]. Apparently, the same features, i.e., enzyme deactivation in the acidic medium and the absence of the branched polymeric protein bonds formation during the analysis, unlike in the case of testing gluten and dough rheology, allowed the SDS sedimentation test to become the main method for assessing the quality of the triticale breeding material in the conditions of Saratov. Thanks to this method, it was possible to identify sources and donors (genotypes that transfer their properties to the offspring in crosses) of grain quality in the collection material, namely ‘Ellada’, ‘Valentin’, Kentavr’, KS 88T-142, TI-17, ADM-9, SV-89229, L-711793, ‘Piligrim’, ‘Modus’, etc., and in the local breeding material. It was also possible to assess grain quality of the promising breeding material as early as starting from F_3_, and to create hybrids promising in terms of grain quality [63].

The state of the carbohydrate–amylase complex is known to play a decisive role in understanding the quality of rye (*Secale cereale* L.) grain [64]. Rye protein forms weak intermolecular bonds, and therefore rye gluten does not wash out under ordinary conditions. At the same time, by forming polymer complexes with carbohydrates, protein participates in the formation of rye dough and plays a role in determining consistency of the caryopsis endosperm. In relatively arid regions of rye cultivation, the problem of its grain quality is no longer limited to the issue of sprouting in the ear [65,66]. In this regard, an attempt to study the consistency of rye grain endosperm by sedimentation of the products from its grinding was of interest.

However, a reservation should be made right away that this required much methodological work, and the initial presumably suitable techniques have undergone significant changes. Nevertheless, it was possible to achieve a stable differentiation of winter rye accessions by using one of the experimental variants of the SDS sedimentation test, when the coefficient of intervarietal variation was 15–29%, and the ranges of variation were 60–30 mL (2001), 33–19 mL (2002), 40–23 mL (2003), and 90–37 mL (2004) (author’s previously unpublished data).

The experiment involved 28 varieties of rye and registration of a whole range of grain quality criteria, traditional and experimental. The results of the experiment demonstrated the main problem of rye breeding for quality, that is, the strongest influence being that of growing conditions. Interrelations between the studied indicators have combined in a complex way over the years and in field replications. At the same time, correlations of the sediment volume with the criteria of amylographic testing were revealed in all the years of the study (0.44*–68**), with the parameters of the bread-making assessment in 2 out of 4 years (0.45*–0.70**), with vitreousness in 2 out of 4 years (0.54**–0.64**), with the water absorption capacity of flour in 2 years (0.51*–0.56**), and with the fluorescence probing data in all the years of the study (0.44*–0.66**). In general, the best ability for stable swelling of whole grain flour was demonstrated in the test by the varieties ‘Falyonskaya’, ‘Chulpan’, ‘Volzhanka’, and ‘Saratovskaya krupnozernaya’. Taking into account the experience of the work with triticale, when some traditional indicators failed in individual years (a situation that can be even more pronounced with rye), it is easy to explain some instability in the relationships over the years, which is also manifested in wheat, although to a much lesser extent. All this creates prospects for the use of the sedimentation test in breeding all these crops.

## 4. Effectiveness of Using Fluorescence Probing for Identifying Genotypes Carrying High Grain Quality Traits in Various Grain Crops

As a research method, fluorescence probing has been widely used to study various biological objects since quite long ago [67,68]. There are works on the use of the method of embryo fluorescent staining in assessing the resistance of wheat grain to sprouting in the ear [69].

Fluorescence probing of a suspension of ground grain products in a weak solution of lactic acid is one of the methods of sedimentation testing [70].

The analysis is based on the registration of differences between the tested samples in the swelling of ground grain particles in the acidic medium and, as a consequence, of differences in the rate of particles sedimentation from the suspension. The registration was done by adding a fluorescent probe to the suspension and measuring the intensity of the suspension fluorescence over time. The results of fluorescence probing were influenced by the availability of the probe binding sites, i.e., of hydrophobic zones in ground grain particles, that is, the “packing density” of protein macromolecules, or the consistency of grain endosperm. The samples with the hardest grain swelled poorly in the short time of the analysis (5 min) and, as a result, had a weaker staining and precipitated faster. In this way, it was possible to identify the potentially strong wheat samples, even among the samples of medium to good quality. The long-term studies have revealed a high differentiating capacity of the method and a fairly stable relationship with traditional criteria for assessing grain quality, i.e., the rheological properties of gluten and dough in spring and winter bread wheat [71], and in spring durum wheat [72]. The specially conducted experiments have proved the genetic dependence of the degree of expression of the parameters of fluorescence probing in these crops [70,73]. Model experiments with grain of other crops, in which the protein complex does not play such a leading role in ensuring the dense consistency of the endosperm as in wheat, also showed the prospects for using the method in breeding for grain quality. The fluorescence probing indicators differentiated the samples of triticale, rye, and millet (*Panicum miliaceum* L.) quite well, and there were interrelations with the traditional assessment criteria. For instance, there was a significant correlation of the fluorescence analysis data with the IDK data (0.68**–0.80**), the falling number (0.54*–0.70**), the amylogram height (0.46*–0.78**), and water absorption capacity of flour (0.46*–0.57**) for winter triticale; with amylographic indicators (0.43*–0.59**), the bread-making quality data (0.43*–0.70**), and with those from the rheological testing of the rye-wheat dough (0.44*–0.74**) for rye; and with the yield of grits from grain grinding, which essentially reflects the consistency of endosperm (0.44*–0.74**), with an increase in the volume of porridge during cooking, and with the weight of the porridge (0.51*–0.53*) for millet (author’s previously unpublished data).

Thus, it was proved that the swelling capacity of the ground grain particles, which underlies the technological cycles of any grain crop processing, can be effectively evaluated using the sedimentation test and fluorescence probing, and have an application beyond various wheat species. Further, it can serve as the phenotypic basis for registering the genetic diversity of agricultural crops in terms of grain quality.

On the whole, the fluorescent probing method, similar to other micro-sedimentation methods, is largely designed for breeding and genetic work, since it reveals the potential of the tested sample in regard to the properties of grain storage polymers. Furthermore, as already mentioned, such methods are highly efficient, enabling the user to evaluate a large amount of experimental material in a short time with minimal consumption. The only limitation to the application of this method that should be considered is the assessment of critically defective grain, the ground grain particles of which have lost their swelling capacity due to degradation. However, such material is quite rare and is easily identified by classical methods of both sedimentation analysis and other methods of grain quality assessment. This also leads to the conclusion that this method is inadvisable to use on the batches of commercial grain intended for further processing, where it is not genetically determined potential that important, but its implementation.

## 5. Development of a Sedimentation Analysis Technique for Identifying the Genetic Diversity in Maize and Grain Sorghum in Terms of Grain Quality (Author’s Previously Unpublished Data)

Sedimentation analysis using experimental techniques was carried out on ground grain products of 34 sorghum (*Sorghum bicolor* (L.) Moench) and 24 maize (*Zea mays* L.) samples harvested in 2019. Swelling of the tested material was carried out both in the acetic acid solution and in the presence of SDS. Significant differences were observed between the samples of sorghum (Table 1) and maize (Table 2) in terms of the sediment volume in two methodological variants of the test. At the same time, it should be noted that the data reproducibility was good in all experimental variants in terms of insignificance of differences between laboratory replications, determined from the F-criterion insignificance. Good differentiation of the samples was also observed for other statistical parameters (limits of variation, coefficient of variation) in both variants of the analysis for both crops. At the same time, the differentiation significantly increased in the variant with SDS.

Of interest is the differentiation of both sorghum and maize samples according to the maximum sediment volume for each crop when comparing the data from two variants of the test. One part of the samples reached a certain sediment volume during sedimentation in acetic acid, and retained it in the case of SDS addition. The second part reached the maximum during sedimentation in acetic acid, but experienced significant destruction of the swelling particles upon the addition of SDS, which was expressed in intensive sedimentation and a significant decrease in the sediment volume. The third part, on the contrary, demonstrated the maximum stability of swelling precisely when SDS was added. This observation shows a reflection of the structural qualitative differences between the tested storage polymers in the sorghum and maize grain.

Due to their structural features, some reach the swelling maximum quickly and show instability under additional physicochemical loads; the others are resistant to an increase in such loads; and still others realize their potential only with additional loads applied (Table 3 and Table 4). Moreover, the obtained data are interesting from the point of view of technological processes of grain processing both in the food and fodder industries. To characterize the stability of the raw matter structure in technological processes, the use of the ratio of the SDS sediment volume to that in acetic acid (SDS/AA) suggests itself. The more this ratio exceeds unity, the greater the structural strength potential of a sample will be.

In our experiment with grain sorghum [74], the SDS/AA ratio turned out to be related to physical properties (extensibility (0.38*–0.50**) and IDK data (0.35*–0.52**) of the mixed samples of gluten, which was washed out from the ground material mixtures of two types: weak wheat + sorghum and strong wheat + sorghum. At the same time, the SDS sediment volume was associated with physical properties of gluten washed out in the presence of strong wheat (0.35*–0.44*), while the sediment volume in acetic acid was done with the strengthening of weak gluten by the addition of products of sorghum grain grinding (−0.44*).

In the experiment with maize [75], the SDS/AA ratio was significantly related to the magnitude of maize material integration into gluten from wheat flour, as was the SDS sediment volume (0.49*) related individually, as well as with the extensibility of gluten from the samples of mixtures with weak wheat (0.48*). The sediment volume in acetic acid was influenced by the biochemical composition of the maize grain. Thus, its volume was positively influenced by the content of nitrogen-free extractive substances (0.42*) and negatively by the content of fat (−0.50*), which, probably, could prevent the hydration of grain storage polymers by forming a hydrophobic layer on the ground particles, which, in its turn, was destroyed by the SDS surfactant in the second variant of sedimentation. Thus, the two-stage sedimentation of the products of grain sorghum and maize grinding can yield exhaustive information about the potential of tested samples during processing, co-processing with wheat included, as well as demonstrate their genotypic diversity concerning these traits.

## 6. Genetic Determination of the Sedimentation Analysis Expression

Many researchers in different countries of the world confirm that the expression of the sedimentation number is genetically determined [51]. Since the 1980s to the present day, a huge amount of material has been accumulated on the molecular labeling of this indicator of wheat grain quality, starting with the electrophoresis of storage proteins.

It was found that the average bread-making quality of wheat cultivars and progeny of the cross ‘Atlas 66’ × ‘Atys’, which possessed subunits 3 and 10, coded for by chromosome 1D, was significantly higher than that of wheat samples possessing subunits 2 and 11, their allelic counterparts [76].

To understand the effect of low-molecular-weight (LMW) glutenin alleles at the Glu-A3 locus on sodium dodecyl sulfate (SDS) sedimentation volume and solvent retention capacity (SRC) values, 244 accessions of Chinese wheat (*Triticum aestivum* L.) mini-core collections were investigated. Seven glutenin alleles at the Glu-A3 locus were confirmed by locus-specific polymerase chain reaction (PCR). SDS sedimentation volume was significantly affected by alleles Glu-A3b and Glu-A3g. Based on total average values, 28 varieties carrying Glu-A3b had significantly higher means of SDS sedimentation volume, whereas 19 varieties carrying Glu-A3g had significantly lower means. Alleles Glu-A3d and Glu-A3f significantly increased only SDS sedimentation volume and sucrose SRC value, respectively. The Glu-A3 LMW glutenin subunit could predict 12.8% of the variance in SDS sedimentation volume [77].

An extended evaluation of promising and commercial varieties of common winter wheat led to a conclusion that the allelic composition of gliadins of these genotypes has a high value: there are no gliadins that negatively affect grain quality tested using SDS sedimentation. That is, genotypes of exactly this type are most commercially successful [78].

The high molecular weight glutenin”subu’Its (HMW-GS) of wheat are major determinants of the viscoelastic properties of gluten and dough. The expression of subunits 1Ax1 or 1Dx5 in transgenic wheat led to corresponding decreases in the proportions of endogenous HMW-GS. HMW-GS 1Ax1 and 1Dx5 had contrasting effects on dough quality determined by the alveograph and sedimentation test. Subunit 1Ax1 increased the tenacity (P), extensibility (L), deformation work (W), and sedimentation value, with the increase being related to the level of expression. In contrast, subunit 1Dx5 led to a smaller increment in the tenacity (P), but to drastic decrease in both extensibility (L), deformation work (W), and the sedimentation value [79].

Special hybridological experiments (including those with interspecific crosses) related to the marking of grain quality traits showed the following.

Out of the six HMW-GS genes, *1Ay* is usually not expressed in bread wheat cultivars. In the current study, an active *1Ay* gene was integrated into two Australian wheat cultivars, ‘Livingston’ and ‘Bonnie Rock’, through conventional backcross approach. Three sister lines at BC4F4 generation for each cross were obtained and underwent a series of quality testing. Zeleny sedimentation value was found to be higher in all three lines of ‘Bonnie Rock’ but only in one of the ‘Livingston’ derivatives. Overall, the integration of *Ay* subunit showed significant positive effects in bread making quality. In general, the overall quality of each *Ay* containing lines was improved. However, for a given line, the extent of improvement in different quality parameters depends on the actual levels of alterations of Glu/Gli ratio and UPP as well as the allelic composition of the other five glutenin genes in the NILs [80].

The advanced backcross QTL (AB-QTL) strategy was utilized to locate quantitative trait loci (QTLs) for baking quality traits in two BC_2_F_3_ populations of winter wheat. The backcrosses are derived from two German winter wheat cultivars, ‘Batis’ and ‘Zentos’, and two synthetic, hexaploid wheat accessions, Syn022 and Syn086. The synthetics originate from hybridizations of wild emmer (*T. turgidum* spp. *dicoccoides*) and *T. tauschii*, rather than from durum wheat and *T. tauschii* and thus allowed us for the first time to test for exotic QTL effects on wheat genomes A and B in addition to genome D. At the QTL on chromosome 4B the exotic allele increased the falling number by 19.6% and at the QTL on chromosome 6D the exotic allele led to an increase of the sedimentation volume by 21.7%. The results indicate that synthetic wheat derived from wild emmer × *T. tauschii* carries favorable QTL alleles for baking quality traits, which might be useful for breeding improved wheat varieties by marker-assisted selection [81].

Bread-making quality in wheat and spelt reflects the combination of several, mostly quantitatively inherited, parameters. The aim was to find molecular markers linked to quantitative trait loci (QTL) for quality parameters. Zeleny sedimentation values (Zel), protein (Prot), kernel hardness (KH) and 1000-kernel weight (TKW) of 226 F_5_ recombinant inbred lines (RILs) from a cross between wheat and spelt were assessed in different environments. The dough properties of 204 RILs were assessed with an alveograph. Based on a genetic map of 187 loci, nine QTL were found for Zel and Prot, explaining 47% and 51% of the phenotypic variance, respectively [82].

Experiments aimed to study genotype-environmental interactions in relation to the degree of expression of various criteria for the wheat grain quality were conducted in several geographical locations in Spain [83] and Latin America [84], and showed the following. Genotypic effects were mainly observed for pigment content and SDS volume [83]. The highly significant correlation between the sedimentation volume and Alveograph *W* confirmed that a simple and fast technique such as the sedimentation volume is a valuable tool to predict the more expensive Alveograph *W* parameter [84]. The influence of the genotype on the SDS sedimentation index expression in spring durum wheat significantly exceeds other factors [85].

Special hybridological experiments for assessing the level and nature of heritability of such traits as the sediment volume and of fluorescent sounding indicators have yielded the below results for progenies and populations.

The Hayman genetic analysis of common spring wheat showed that the fluorescence probing criteria recommended for testing gluten quality in early generations are controlled (with a few exceptions) by an additive-dominant gene system. Their inheritance mainly follows the incomplete dominance pattern, and the increase in the quantitative expression of most of them is associated with recessive genes. Early selection of genotypes with high gluten quality according to these characters is quite possible [70]. The rate of SDS sedimentation in durum spring wheat is controlled by genes with epistatic, additive and dominant effects [85].

The fluorescent analysis indicators for populations of common spring wheat, common winter wheat, and durum spring wheat show a high level of heritability with positive selections of various intensity [73].

Protein and gluten content, gluten quality and sedimentation index follow different patterns of inheritance in common winter wheat. In segregating progenies, the portion of combinations with positive dominance and intermediate inheritance is the greatest. The coefficient of heritability is the highest for the “sedimentation” trait and is less significant for “gluten quality”. The sedimentation index is one of the most reliable marker traits in breeding for the increased flour strength [86].

High heritability coupled with genetic advance was observed for SDS sedimentation in combined analysis. This implies the potential of improving wheat for end product use quality through direct selection [87]. It is emphasized that the sedimentation analysis is the most objective criterion for the breeding-oriented evaluation already in early generations, which allows both positive and negative selection with regard to grain quality within a wide diversity of genotypes [88].

A special mentioning is deserved by the extensive work performed by Ruan Y et al. (2020) on the molecular genetics of durum spring wheat in relation to grain quality, which reports as follows: “To characterize the quantitative trait loci (QTL) controlling gluten strength in Canadian durum wheat cultivars, a population of 162 doubled haploid (DH) lines segregating for gluten strength and derived from cv. ‘Pelissier’ × cv. ‘Strongfield’ was used in this study. The DH lines, parents, and controls were grown in 3 years and two seeding dates in each year and gluten strength of grain samples was measured by sodium dodecyl sulfate (SDS)-sedimentation volume (SV). With a genetic map created by genotyping the DH lines using the Illumina Infinium iSelect Wheat 90K SNP (single nucleotide polymorphism) chip, QTL contributing to gluten strength were detected on chromosome 1A, 1B, 2B, and 3A. Two major and stable QTL detected on chromosome 1A (*QGlu.spa-1A*) and 1B (*QGlu.spa-1B.1*) explaining 13.7–18.7% and 25.4–40.1% of the gluten strength variability respectively were consistently detected over 3 years, with the trait increasing alleles derived from ‘Strongfield’. Putative candidate genes underlying the major QTL were identified. Two novel minor QTL (*QGlu.spa-3A.1* and *QGlu.spa-3A.2*) with the trait increasing allele derived from ‘Pelissier’ were mapped on chromosome 3A explaining up to 8.9% of the phenotypic variance; another three minor QTL (*QGlu.spa-2B.1*, *QGlu.spa-2B.2*, and *QGlu.spa-2B.3*) located on chromosome 2B explained up to 8.7% of the phenotypic variance with the trait increasing allele derived from ‘Pelissier’. *QGlu.spa-2B.1* is a new QTL and has not been reported in the literature. Multi-environment analysis revealed genetic (QTL) × environment interaction due to the difference of effect in magnitude rather than the direction of the QTL. In total, 11 pairs of digenic epistatic QTL were identified, with an epistatic effect between the two major QTL of *QGlu.spa-1A* and *QGlu.spa-1B.1* detected in four out of six environments. The peak SNPs and SNPs flanking the QTL interval of *QGlu.spa-1A* and *QGlu.spa-1B.1* were converted to Kompetitive Allele Specific PCR (KASP) markers, which can be deployed in marker-assisted breeding to increase the efficiency and accuracy of phenotypic selection for gluten strength in durum wheat” [89].

## 7. Directions for Further Research

Since, as already noted, the sedimentation method of grain quality assessment can reveal the genetically determined potential of the tested sample, the prospects for its widespread use in breeding and genetic programs are indubitable both for the identification of valuable genotypes in regard to grain quality in a breeding process and for studying the inheritance mechanisms of valuable properties of grain storage polymers by methods of molecular genetics.

Most techniques of grain crop processing for food purposes are based on the stable swelling capacity of ground grain products. At the same time, different processing cycles can impose various requirements on the properties of raw materials, including the types and dynamics of swelling. Thus, one of the possible directions of developing the sedimentation method for its further application is the selection of genotypes capable of producing raw materials that meet these requirements.

Moreover, as was shown in this work, methodological research aiming at the expansion of the list of research objects by using the sedimentation method is of great interest.

## 8. Conclusions

In general, the data obtained and the material accumulated by various researchers indicate that a modification of the sedimentation method, correctly chosen for specific goals and objectives, solves the problem of assessing grain quality in breeding samples starting from early progenies. The use of sodium dodecyl sulfate (SDS) reveals the potential of strong protein formations: the inclusion of complex conglomerates in the hydrophobic zones ensures their uniform and maximum swelling in the acidic environment. In other cases, it stabilizes the structure of weak, i.e., simpler polymer formations when they dominate in the ground grain products of certain crops. The high level of genetic determination of the sedimentation number expression distinguishes it from other criteria for assessing grain quality.

All these circumstances make sedimentation testing the leading or most promising method for assessing grain quality when the breeding of a broad range of grain crops is carried out.

## Figures and Tables

**Table 1 plants-11-01640-t001:** Statistical differentiation of grain sorghum samples by the sediment volume.

**Acetic Acid Variant**	
Range of variation, mL	24.0–47.5
Average per test, mL	35.6
Coefficient of variation	14.5
F-criterion	7.32 *
Smallest significant difference	5.5
**SDS Variant**	
Range of variation, mL	15.0–62.0
Average per test, mL	36.9
Coefficient of variation	25.5
F-criterion	21.51 *
Smallest significant difference	5.8

*—reliability of F-criterion.

**Table 2 plants-11-01640-t002:** Statistical differentiation of maize samples by the sediment volume.

**Acetic Acid Variant**	
Range of variation, mL	18.5–53.5
Average per test, mL	32.5
Coefficient of variation	23.6
F-criterion	17.58 *
Smallest significant difference	5.4
**SDS Variant**	
Range of variation, mL	15.0–70.0
Average per test, mL	38.6
Coefficient of variation	45.7
F-criterion	69.6 *
Smallest significant difference	6.2

*—reliability of F-criterion.

**Table 3 plants-11-01640-t003:** Differentiation of grain sorghum genotypes according to the type of milled grain swelling.

Swelling 1st Type		Swelling 2nd Type		Swelling 3rd Type	
Accession	SDS/AA	Accession	SDS/AA	Accession	SDS/AA
‘Kamelik’	0.34	‘Oniks’	1.00	‘Partizan’	1.83
‘Zhemchug’	0.55	‘Avans’	1.01	‘Topaz’	1.55
‘Perspektivnyi 1’	0.68	‘Bakalavr’	1.02	‘Granat’	1.48
‘Azart’	0.71	‘Pishchevoye 614’	1.03	‘Pishchevoye 35’	1.47
‘Kokhalong’	0.78	‘Korall’	0.97	‘Infinity’	1.43
‘Ogonyok’	0.79	‘Lokus’	0.97	‘Volzhskoye 44’	1.43
‘Magistr’	0.83	‘Kremovoye’	1.04	‘Assistant’	1.32
‘Geleofor’	0.84	‘Start’	1.04	‘Meteor’	1.26
		‘Fakel’	1.05	‘Sarmat’	1.20
		‘Studenets’	0.94	‘Merkuriy’	1.19
		‘Zenit’	0.91	‘Vostorg’	1.17
		‘Volzhskoye 4v’	0.91	‘Volzhskoye 4’	1.15
		‘Volzhskoye 615’	1.12		

**Table 4 plants-11-01640-t004:** Differentiation of maize genotypes according to the type of milled grain swelling.

Swelling 1st Type		Swelling 2nd Type		Swelling 3rd Type	
Accession	SDS/AA	Accession	SDS/AA	Accession	SDS/AA
‘Dublyor’	0.36	Mestnaya (Kazakhstan)	1.0	‘HLG1380’	2.09
‘Klinok’	0.37	‘HL926’	1.03	‘YUV3zM’	2.08
‘In 178-2’	0.72	‘ND245’	1.09	‘RNIISK’	2.00
‘Bankutskaya’	0.47	No 515	1.12	‘Zarya’	1.93
‘Stimul’	0.67	‘Tsukerka’	1.14	‘Raduga’	1.71
‘SM105’	0.80			‘Avrora’	1.67
‘Matador’	0.88			‘HLG912’	1.54
				‘HLG1003’	1.53
				‘Jaune gros’	1.44
				‘Choriella’	1.43
				‘Zabava’	1.30
				‘F674’	1.19

## Data Availability

Not applicable.

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
