# Peer review of "Effectiveness of and Perspectives for the Sedimentation Analysis Method in Grain Quality Evaluation in Various Cereal Crops for Breeding Purposes"

_plants, 2022, doi:10.3390/plants11131640_

Round 1
Reviewer 1 Report
I have reviewed ‘effectiveness of and perspectives for the sedimentation analysis method in grain quality evaluation in various cereal crops for breeding purposes’. Overall the paper is well written and provide sufficient data, discussion, and literature review. The results could provide valuable information to the researchers in breeding area. Some other comments:
1. Abstract, please add the major conclusions of this paper.
2. Line 50: formatting issue.
3. Line 153-156: please add reference.
4. Line 169, 178: punctuation problem.
5. Line 169-178: please indicate what does * and ** mean. In addition, please add reference.
6. The author included a lot of previous unpublished data, it is suggested the author also provide the methods of how the previous data were obtained in the supplementary document.
7. Line 400, what does XX mean?
8. Please expand the section 6: genetic determination of the sedimentation analysis expression.
9. Please add a future direction section.
10. Patents should not be a section of the manuscript, please put in other parts.
Author Response
Thank you for your time and appreciation of my work. We have answered each of your points below.
1) Abstract, please add the major conclusions of this paper.
2) Строка 50: проблема форматирования.
3) Строка 153-156: пожалуйста, добавьте ссылку.
4) Строки 169, 178: пунктуационная проблема.
5) Строка 169-178: пожалуйста, укажите, что означают * и **. Кроме того, пожалуйста, добавьте ссылку.
Ответ: исправлено (1-5)
6) Автор включил много предыдущих неопубликованных данных, предлагается автору также привести методы получения предыдущих данных в дополнительном документе.
Ответ: В статью добавлен дополнительный справочный материал с перечнем используемых методов, описание которых не вошло в основной текст статьи.
7) Строка 400, что означает XX?
Ответ: исправлено
8) Пожалуйста, расширьте раздел 6: генетическое определение экспрессии седиментационного анализа.
Ответ: исправлено
9) Пожалуйста, добавьте раздел будущего направления.
Ответ: исправлено
10) Патенты не должны быть разделом рукописи, пожалуйста, поместите в другие части.
Ответ: исправлено
С уважением,
д-р Кибкало
Reviewer 2 Report
The authors systemically introduce the sedimentation testing involved in assessing a broad range of grain crop quality in this manuscript. It is well-written and provides the most updated literature. Still, it has a few moments to correct, including:
1) The authors should better provide a flow chart to show the working principle of this method, not only some comparison tables.
2) The author should describe the fluorescent probing method's advantages, disadvantages, and applications and a two-stage sedimentation procedure, two parts of the sedimentation testing.
3) The font of the reference part should adjust for the whole article.
Author Response
Thank you for your time and appreciation of my work. We have answered each of your points below.
1) The authors should better provide a flow chart to show the working principle of this method, not only some comparison tables.
Response: Additional reference material has been added to the article listing the methods used, the description of which was not included in the main text of the article. The procedure of sedimentation analysis, including a two-stage version, is also outlined there.
2) The author should describe the fluorescent probing method's advantages, disadvantages, and applications and a two-stage sedimentation procedure, two parts of the sedimentation testing.
Response: corrected
The section on the fluorescence analysis has been expanded. Its advantages and disadvantages are described in more detail.
3) The font of the reference part should adjust for the whole article.
Response: corrected
Best regards,
Dr. Kibkalo
Reviewer 3 Report
The manuscript brings not new elements to existing knowledge on topic
The manuscript is not well prepared professionally. It includes not a well-crafted abstract and and not include hypothesis. The aim is not clearly defined. The discussion of the results is not well prepared. The conclusions are not well-defined. The illustrative material is not appropriate.
In my opinion, the manuscript not suitable for publication in this journal.
Author Response
Thank you for the time devoted to my work.
The purpose of the manuscript was to summarize and analyze the accumulated data obtained during the methodological research aimed at the identification of genotypic diversity by grain quality using the sedimentation method. Original observations include the changes in the variants of the sedimentation analysis depending on the type of wheat, its place of cultivation, as well as the application of the SDS micro-sedimentation method to assess grain properties of other cereals – rye, triticale, corn, sorghum. Also, the data on the use of the fluorescence analysis on various cereals, which can be regarded as a kind of the sedimentation method, is provided.
The article largely presents the author's long-term data on several experiments. Each of these experiments may form the basis for an experimental article, and a considerable part of this data was separately published before. Therefore, to summarize the material in the direction indicated, the form of a review article was chosen, although some of the experimental material had not been previously published. Since the form of the submitted manuscript is a review, it does not have the structure typical of an experimental article.
Given that the previously unpublished experimental data is of particular interest, additional information material has been added to the article. Some changes have also been made to the manuscript, including the abstract.
Sincerely,
Dr. Kibkalo
Round 2
Reviewer 3 Report
The author made great efforts to improve manuscript. The paper is now ready for publication